# The gene *"degrees of kevin bacon"* (*dokb*) regulates a social network behaviour in *Drosophila melanogaster*

Rebecca Rooke[1,2], Joshua J. Krupp[1], Amara Rasool[1,2], Mireille Golemiec[1], Megan Stewart[1], Jonathan Schneider [1] & Joel D. Levine [1,2] ✉

Social networks are a mathematical representation of interactions among individuals which are prevalent across various animal species. Studies of human populations have shown the breadth of what can spread throughout a social network: obesity, smoking cessation, happiness, drug use and divorce. 'Betweenness centrality' is a key property of social networks that indicates an individual's importance in facilitating communication and cohesion within the network. Heritability of betweenness centrality has been suggested in several species, however the genetic regulation of this property remains enigmatic. Here, we demonstrate that the gene *CG14109*, referred to as *degrees of kevin bacon* (*dokb*), influences betweenness centrality in *Drosophila melanogaster*. We identify strain-specific alleles of *dokb* with distinct amino acid sequences and when the *dokb* allele is exchanged between strains, flies exhibit the betweenness centrality pattern dictated by the donor allele. By inserting a GAL4 reporter into the *dokb* locus, we confirm that *dokb* is expressed in the central nervous system. These findings define a novel genetic entry point to study social network structure and thereby establish gene-to-social structure relationships. While *dokb* sequence homology is exclusive to Diptera, we anticipate that *dokb*-associated molecular pathways could unveil convergent neural mechanisms of social behaviour that apply in diverse animal species.

Although the mention of social networks often brings to mind digital social media platforms like Facebook and other online communities, the use of social networks to analytically study group behaviour originated in the field of sociology in the 1930s[1]. Since its inception, social networks have been studied in various animals, from fruit flies[2] to elephants[3,4]. Despite how common it is for animals to form social groups and interact within them, genetic contributions to the structure of group behaviour are still poorly understood. Here, we use the common fruit fly, *Drosophila melanogaster*, to identify a gene responsible for regulating a social network property, betweenness centrality.

Betweenness centrality, like other measures of centrality, quantifies the importance of individuals in facilitating interactions within a social network and is defined as the number of shortest paths that traverse a node/individual[5]. In some animals, such as dolphins and rock hyraxes, high measures of centrality correlate to better health outcomes[6,7]. An individual with high betweenness centrality is thought to be important for cohesion and communication relay throughout a group and can be loosely thought of as a "gatekeeper" for the network (Fig. 1). Depending on what is spreading throughout a social network, having high betweenness centrality may underlie beneficial or detrimental outcomes. For example, having high betweenness centrality in

[1]Department of Biology, University of Toronto at Mississauga, 3359 Mississauga Rd. North, Mississauga, ON L5L 1C6, Canada. [2]Department of Ecology and Evolutionary Biology, University of Toronto, 25 Willcocks St., Toronto, ON M5S 3B2, Canada. ✉e-mail: joel.levine@utoronto.ca

**Fig. 1 | Social network structure and betweenness centrality.** Circles represent nodes (individuals), and the arrowed connections indicate interactions between nodes. Numbers indicate the degree (total number of interactions) for each node. Anything flowing through the network from the yellow nodes on the left to the yellow nodes on the right (or vice versa) must traverse through the blue node. Thus, the blue node has a higher betweenness centrality than the yellow nodes, as the blue node acts as a 'gatekeeper' between the left side and right side of this network.

a scientific research network increases potential collaborators[8], but in a disease network, it may increase the chance of HIV infection[9]. Studies have shown that betweenness centrality is likely heritable in several species, including marmots[10], macaques[11], fruit flies[12] and humans[13].

Betweenness centrality is one structural feature of a social network, although there are many others[14]. The betweenness centrality of a group guides and constrains how the group may function. For example, the betweenness centrality of a group can influence the spread of a disease, the spread of information or the distribution of food. Similar to mating, courtship and aggression, betweenness centrality only manifests in a social setting. Here, we use *Drosophila melanogaster* social networks (Supplementary Fig. 1) to identify and manipulate a single gene that regulates betweenness centrality (BC).

## Results

### Mapping a locus responsible for regulating BC in *Drosophila*

To identify a gene that regulates betweenness centrality, we began by using an adapted recombinant mapping technique[15] and exploited the observation that betweenness centrality is higher in our Canton-S (CS) strain compared to our Oregon-R (OR) strain of *Drosophila melanogaster*[16] (Supplementary Fig. 2). The strain-specific phenotype and mapping method were robust and allowed us to map a betweenness centrality locus within a 474 kb region on the 3L chromosome (Fig. 2a, b; Supplementary Figs. 2 and 3). We identified two genes within this locus that are differentially expressed in the central nervous systems of CS and OR flies: *Nplp2* and *CG14109* (Supplementary Table 1).

### Testing candidate genes, *Nplp2* and *CG14109*, for the regulation of BC

Next, we asked whether either of these genes influence betweenness centrality. To address this question, we knocked down the expression of *Nplp2* and *CG14109* using the GAL4-UAS system and eliminated the expression of *CG14109* by creating a null mutant using CRISPR/Cas9. Both *Nplp2* and *CG14109* RNAi lines and the *CG14109* null mutant had a strong knockdown efficacy (see below). We show that knocking down *Nplp2* expression did not affect betweenness centrality, but knocking down or knocking out expression of *CG14109* resulted in networks with lower betweenness centrality compared to wild-type controls (see below).

Next, we asked whether *CG14109* is responsible for the strain-based difference in betweenness centrality. This gene differs by seven nucleotides between our CS and OR strains, one of which results in a predicted amino acid difference ($129_{Ala \to Glu}$; Fig. 2c, d; Supplementary Fig. 5). However, there are no nucleotide differences in the upstream region of the gene. We used CRISPR/Cas9 to insert the CS *CG14109* allele into an OR fly and vice versa (Fig. 2c, d). Networks formed by these flies show that betweenness centrality correlates to the *CG14109* donor allele and not the genetic background (Fig. 2e): just as CS wild-

type flies form networks with higher betweenness centrality than OR wild-type flies (Fig. 2e; Supplementary Fig. 2; Supplementary Fig. 3), flies with a CS *CG14109* allele (*CG14109^{+1}*) in an otherwise OR background form networks with higher betweenness centrality than flies with an OR *CG14109* allele (*CG14109^{+2}*) in a CS background. These data show that the *CG14109* allele rescues the robust strain-based difference observed between our CS and OR strains. Other social behaviours correspond most closely to the genetic background of these lines and not the *CG14109* donor allele (Fig. 3a, c, e), although some social behaviour measures did not correspond to either the *CG14109* allele or the genetic background (Fig. 3b, d). We conclude that *CG14109* regulates betweenness centrality, a metric of group behaviour. We named this gene *degrees of kevin bacon* (*dokb*) after the parlor game "Six Degrees of Kevin Bacon" where participants choose an arbitrary Hollywood actor and determine the shortest path that connects that individual to Kevin Bacon *via* film roles.

### *dokb* expression correlates with social network BC

Next, we asked whether *dokb* RNA expression correlates with the betweenness centrality phenotype. To do this, we manipulated *dokb* expression using mutant and transgenic lines and subsequently measured the betweenness centrality of their networks. Knocking out and knocking down *dokb* expression reduced betweenness centrality (Fig. 4a–c; Supplementary Fig. 6a, b). Additionally, we found that when the *dokb^{+1}* allele was inserted into an OR background, *dokb* was expressed at the same levels as in CS flies, and when the *dokb^{+2}* allele was inserted into a CS background, it was expressed at the same levels as in OR flies (Fig. 4d; Supplementary Fig. 6c). The fact that the upstream region of *dokb* is identical in both strains suggests that differential *dokb* expression is a result from regulatory elements within the gene itself. Moreover, we show that *dokb* expression did not decrease in *Nplp2* knockdown flies, and the behavioural phenotype did not differ from controls (Fig. 4e; Supplementary Fig. 6d). Taken together, these data demonstrate that *dokb* RNA expression correlates with betweenness centrality and supports our conclusion that *dokb* regulates the betweenness centrality phenotype in fly social networks.

### Deleting *dokb* changes hydrocarbon profiles in a strain-specific way

Given that betweenness centrality is a group-level behaviour and social interactions among *Drosophila* conspecifics are primarily mediated by detecting hydrocarbons[17–19], we next investigated whether deleting *dokb* leads to modified hydrocarbon profiles. In a CS background, *dokb* nulls (*dokb^{n1}*) showed an increase in alkanes, methyl alkanes and total hydrocarbon amounts, but there was no effect of deleting *dokb* on total alkene production (Supplementary Fig. 7a–d). In an OR background, *dokb* null flies (*dokb^{n2}*) had reduced alkenes, alkanes and total hydrocarbon amounts and increased methyl alkanes (Supplementary Fig. 7a–d). *dokb* nulls of either strain showed no effect on cVA production, a compound previously shown to cause aggregation and dispersal in *Drosophila*[20–24] (Supplementary Fig. 7e). To investigate whether *dokb* can influence the production of hydrocarbons directly, we next asked whether *dokb* is expressed in the oenocytes, the cells that synthesize cuticular hydrocarbons in *Drosophila* and other insects, using qPCR and a *dokb* GFP reporter. The oenocytes expressed extremely low levels of *dokb* with most of our replicates failing to detect any *dokb* at all (Supplementary Table 2). No GFP was detected in the oenocytes (Fig. 5d); however, GFP was detected in adult somatic muscle tissue including, but not limited to, the dorsal lateral muscles (Fig. 5a, b), the alary muscles (Fig. 5c, d) and larval body wall muscles (Fig. 5e). Taken together, our data show that deleting *dokb* results in changes to hydrocarbon profiles but *dokb* is not reliably detected in the oenocytes, suggesting it does not directly affect hydrocarbon production. These results are consistent with previous findings, that group composition can affect hydrocarbon profiles and gene

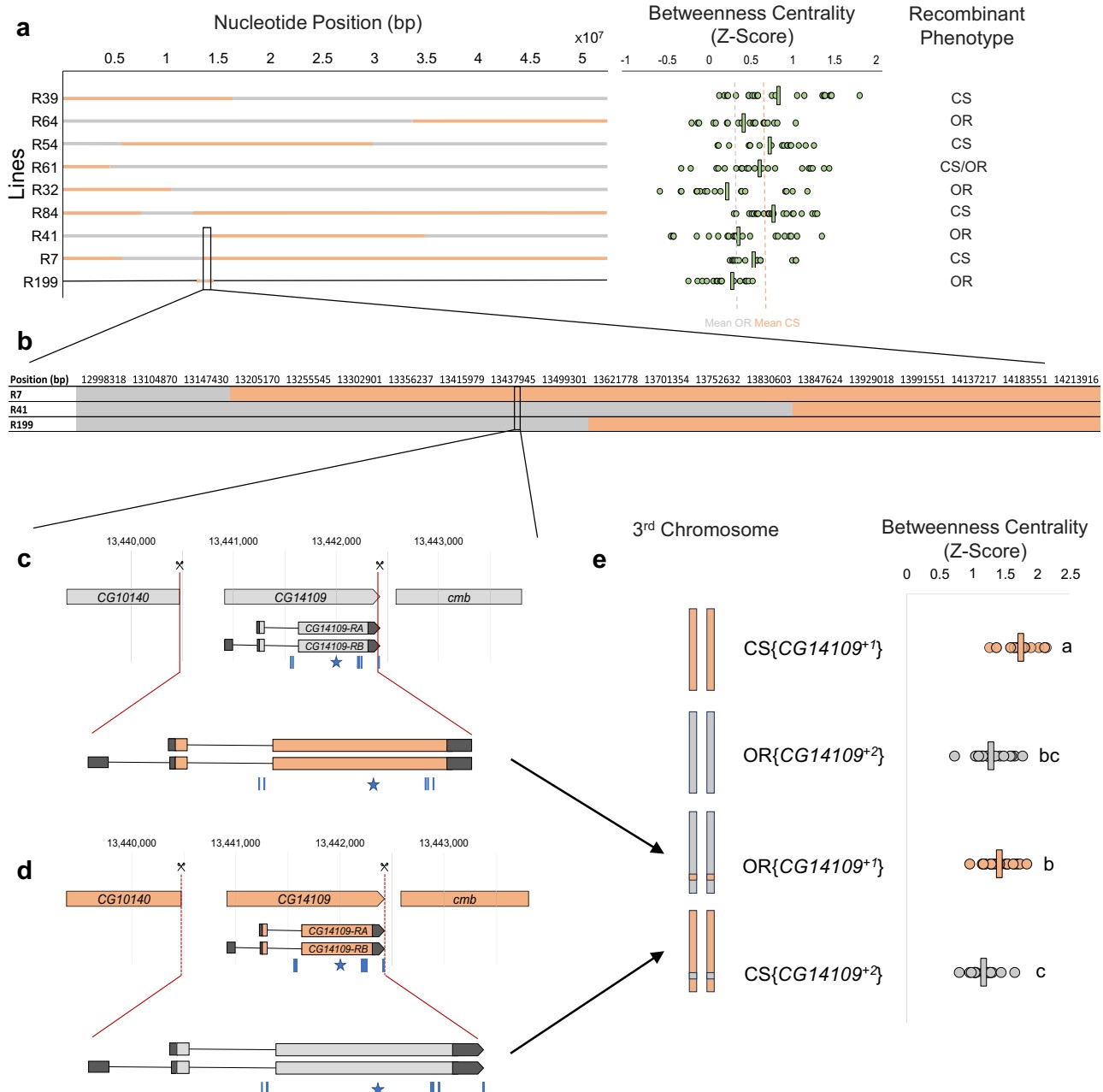

**Fig. 2 | Mapping the gene that regulates betweenness centrality. a** Left: Visual representation of recombination events on the 3rd chromosome generated through recombination of CS and OR strains of *Drosophila* as determined by SNP genotyping. OR regions are displayed in grey, CS regions are displayed in orange, unsequenced regions are displayed as a thin black line. Nucleotide position (bp) of chromosome 3 is on the *x*-axis. Middle: A summary of the betweenness centrality that was calculated from the social networks of different groups of flies. Each dot represents the average betweenness centrality from a network formed by a distinct group of 12 flies. Averages for each genotype are shown by a vertical line. The average betweenness centrality of all CS and OR controls used in the recombinant mapping experiments are displayed as orange and grey dashed lines, respectively. Right: The phenotype of the recombinant lines was determined by comparing the recombinant lines to wild-type controls. See Supplementary Fig. 2 for sample sizes and statistics for all recombinant lines. **b** The black box represents the region housing gene(s) responsible for regulating betweenness centrality. A higher resolution genotyping using PCR was used to determine where recombination occurred in lines R7, R41 and R199. **c** and **d** Pictures representing the creation of *CG14109* swap lines. Blue lines = the approximate position of the CS/OR SNPs. Blue star = approximate position of CS/OR SNP that results in a $129_{Ala \rightarrow Glu}$ missense mutation. **c** OR fly with a CS allele of the *CG14109* gene (referred to as OR{*CG14109*^{+1}}). **d** CS fly with an OR *CG14109* allele (referred to as CS{*CG14109*^{+2}}). **e** Betweenness centrality of CS wild-type (CS{*CG14109*^{+1}}) and OR wild-type (OR{*CG14109*^{+2}}) lines with the two swap lines. Depictions of each line's third chromosome are on the left. Each dot represents the average betweenness centrality from a network formed by a distinct group of 12 flies. Averages for each genotype are shown by a vertical line. Statistical significance is indicated by letters and was determined by a one-way ANOVA, followed by a Tukey-Kramer post hoc test. $F_{(3,77)} = 23.71$, $p = 7.38 \times 10^{-11}$. CS{*CG14109*^{+1}}: $n = 18$, OR{*CG14109*^{+2}}: $n = 21$, OR{*CG14109*^{+1}}: $n = 20$, CS{*CG14109*^{+2}}: $n = 19$. Source data are provided in the Source Data file.

expression independent of genotype[18,25]. Thus, we favour the hypothesis that changes in hydrocarbon profiles in *dokb* null mutants are a consequence of their social experience; i.e. differences in their social network structure.

## *dokb* is expressed in larval and adult muscle tissue and the CNS

To determine where *dokb* is expressed in the *Drosophila* central nervous system (CNS), we crossed our *dokb^{n2}*-GAL4 line to a fluorescent reporter. *dokb* expression in the larval central brain begins in

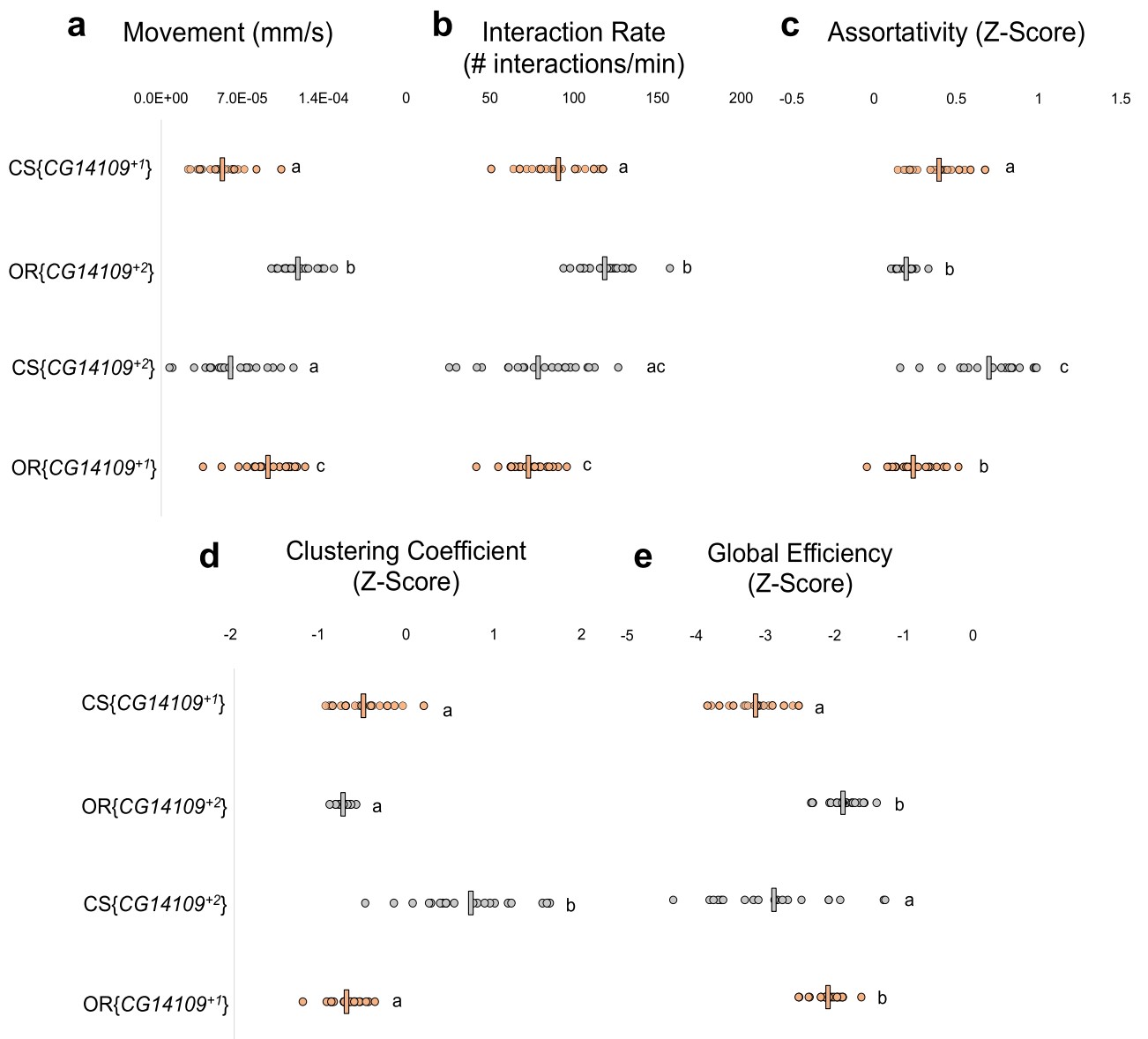

**Fig. 3 | Behavioural properties of CS{*CG14109*⁺²} and OR{*CG14109*⁺¹} flies and their controls.** Each dot represents the average measurement from a network formed by a distinct group of 12 flies. Vertical lines indicate the average measurement for a given genotype. Letters indicate statistical significance, as determined by a one-way ANOVA followed by a Tukey–Kramer post hoc test. **a** Average movement of flies for each genotype. $F_{(3,82)} = 39.10$, $p = 1.36 \times 10^{-15}$. CS{*CG14109*⁺¹}: $n = 21$, OR{*CG14109*⁺²}: $n = 21$, CS{*CG14109*⁺²}: $n = 21$, OR{*CG14109*⁺¹}: $n = 20$. **b** Average interaction rate of flies for each genotype. $F_{(3,83)} = 22.28$, $p = 1.39 \times 10^{-10}$. CS{*CG14109*⁺¹}: $n = 22$, OR{*CG14109*⁺²}: $n = 21$, CS{*CG14109*⁺²}: $n = 21$, OR{*CG14109*⁺¹}: $n = 20$. **c** Average assortativity of flies for each genotype. $F_{(3,79)} = 39.89$, $p = 1.36 \times 10^{-15}$. CS{*CG14109*⁺¹}: $n = 22$, OR{*CG14109*⁺²}: $n = 18$, CS{*CG14109*⁺²}: $n = 19$, OR{*CG14109*⁺¹}: $n = 21$. **d** Average clustering coefficient of flies for each genotype. $F_{(3,81)} = 79.13$, $p = 1.34 \times 10^{-23}$. CS{*CG14109*⁺¹}: $n = 22$, OR{*CG14109*⁺²}: $n = 18$, CS{*CG14109*⁺²}: $n = 21$, OR{*CG14109*⁺¹}: $n = 21$. **e** Average global efficiency of flies for each genotype. $F_{(3,80)} = 31.71$, $p = 1.34 \times 10^{-23}$. CS{*CG14109*⁺¹}: $n = 21$, OR{*CG14109*⁺²}: $n = 21$, CS{*CG14109*⁺²}: $n = 19$, OR{*CG14109*⁺¹}: $n = 20$. Source data are provided in the Source Data file.

the 2nd instar larval stage and persists into adulthood (Supplementary Fig. 8, Fig. 6). In the 3rd instar larval CNS, GFP was detected in the mushroom body calyces, the ventral nerve cord, the eye/antennal disc and leg discs (Supplementary Fig. 9). In the adult male CNS, GFP was detected in the gamma lobes and calyces of the mushroom bodies, as well as the suboesophageal ganglion and ventral nerve cord (Fig. 6). The *Drosophila* mushroom bodies have been shown to play a critical role in olfactory learning and memory[26–29] and are connected to several primary sensory centres, including the olfactory antennal lobe, visual optic lobe and the gustatory suboesophageal zone[30]. The gamma lobes of the mushroom bodies have been shown to play a role in social attraction and in forming short-term memories in *Drosophila*[31–35] (Fig. 6a–c, f, g). In

the ventral nerve cord, GFP is expressed in the ventral and intermediate regions of the prothoracic, mesothoracic and metathoracic neuromeres[36], which are regions innervated by neurons associated with the legs[37] and regions serving to link legs and wing control[38], respectively. GFP expression is also evident in the accessory mesothoracic neuropil, which is associated with sensory afferents from the wing and notum[36,39] (Fig. 6a–c). Taken together, we demonstrate that *dokb* expression begins in the larval stage and occurs in tissue-specific regions within the larval and adult central brain and ventral nerve cord. In adults, *dokb* expression patterns coincide with tissues related to olfactory learning, locomotion and motor control. Given that *dokb* regulates a complex group-level behaviour, it is noteworthy that this gene's expression pattern suggests a potential

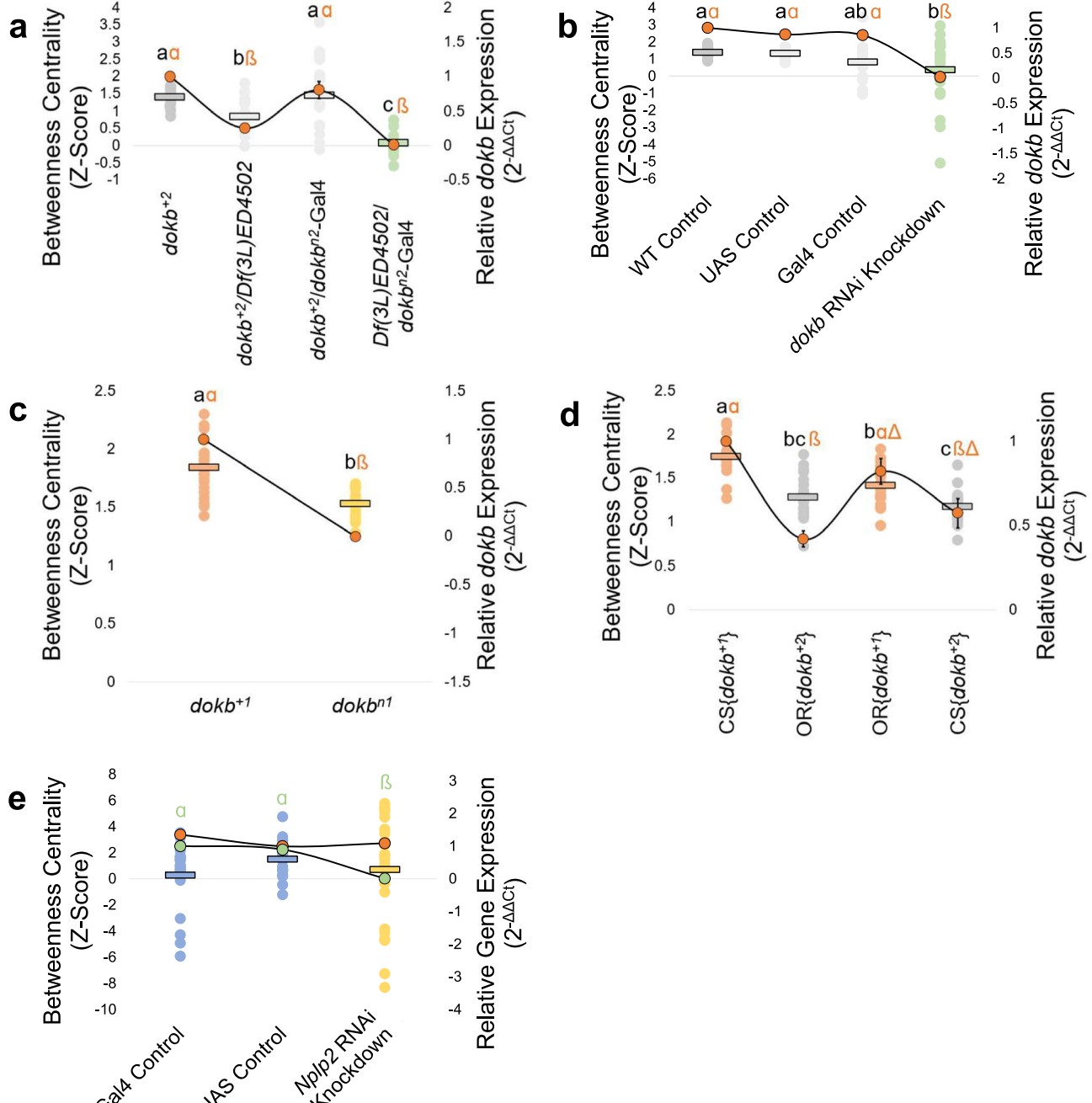

**Fig. 4 | *dokb* RNA expression and betweenness centrality across genetic lines.**
Each dot (no outline) represents the average BC from a network formed by a distinct group of 12 flies. Average BC for each genotype is shown by horizontal lines. Outlined dots connected by a black line represent the average *dokb* (orange) or *Nplp2* (green) expression. A two-tailed *t*-test was used for all statistical analyses for graph **c**. For all other graphs, the statistical significance of BC is indicated by black letters and was determined by a one-way ANOVA, followed by a Tukey–Kramer post hoc test. Statistical significance of average gene expression is indicated by Greek letters as determined by a one-way ANOVA, followed by a Tukey–Kramer post hoc test. Error bars indicate ±SE for RNA expression. **a** *dokb* expression and BC of genotypes containing two copies of *dokb* (*dokb*+2), one copy of *dokb* (*dokb*+2/Df(3 L) ED4502 and *dokb*+2/ *dokb*n2-GAL4) and no copies of *dokb* (Df(3L)ED4502/ *dokb*n2-GAL4). Behaviour: *dokb*+2: *n* = 22; *dokb*+2/Df(3 L)ED4502: *n* = 19; *dokb*+2/ *dokb*n2-GAL4: *n* = 18; Df(3L)ED4502/ *dokb*n2-GAL4: *n* = 12. $F_{(3,70)} = 17.41$, $p = 1.83 \times 10^{-8}$. *dokb* RNA Expression: *n* = 4 for all genotypes. $F_{(3,15)} = 47.19$, $p = 6.52 \times 10^{-7}$. **b** Knocking down

*dokb* expression decreases BC. Behaviour: WT Control: *n* = 21, UAS Control: *n* = 20, GAL4 Control: *n* = 19, *dokb* RNAi Knockdown: *n* = 22. $F_{(3,81)} = 3.81$, $p = 1.73 \times 10^{-6}$. *dokb* RNA Expression: *n* = 3 for all genotypes. $F_{(3,11)} = 89.16$, $p = 1.73 \times 10^{-6}$. **c** *dokb*n1 flies have no *dokb* expression and a decrease in BC. Behaviour: *dokb*+1: *n* = 22; *dokb*n1: *n* = 18. $t_{17} = 5.74$, $p = 2.37 \times 10^{-5}$. dokb RNA Expression: *n* = 4 for each genotype. $t_3 = 3.01 \times 103$, $p = 8.01 \times 10^{-11}$. **d** *dokb* expression correlates to the *dokb* allele. Behaviour: data is the same as in Fig. 2e (see figure legend). *dokb* RNA Expression: *n* = 4 for each genotype. $F_{(3,15)} = 16.86$, $p = 1.33 \times 10^{-4}$. **e** BC and *dokb* RNA expression levels (orange dots with outlines connected by a black line) remain stable in *Nplp2* RNAi knockdown flies and controls. *Nplp2* expression is reduced in *Nplp2* RNAi knockdown flies (green dots with outline connected by a black line). Behaviour: GAL4 Control: *n* = 21; UAS Control: *n* = 20; Nplp2 Knockdown: *n* = 24. Data is n.s ($p = 0.42$). *dokb* RNA Expression: *n* = 3 for each genotype. Data is n.s. ($p = 0.23$). *Nplp2* RNA Expression: *n* = 3 for each genotype. $F_{(2,8)} = 103$, p = $2.21 \times 10^{-5}$. Source data are provided in the Source Data file.

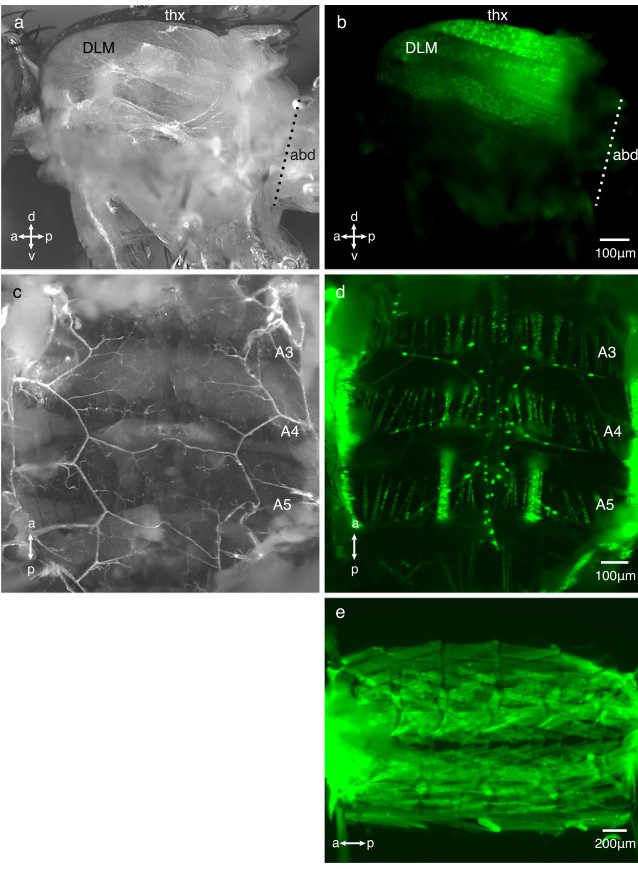

**Fig. 5 | Expression of *dokb* using dokb^{n2}-GAL4 line and a GFP reporter in adult and larval muscle tissue.** *dokb^{n2}*-GAL4 expression patterns reported by a UAS-GFP reporter (green). Scale bars are shown at the bottom. **a** and **b** Sagittal section through the adult thorax exposing the indirect flight muscles. Reporter: UAS-StingerII. **a** Brightfield image showing the dorsal lateral muscle (DLM). thx = thorax; abd = abdomen. **b** Fluorescent image showing the DLM. The same preparation is shown in **a** and **b**. Other muscles of the thorax and legs were also labelled but are out of the plane of focus. **c** and **d** Fillet preparation of the adult abdomen. **c** Brightfield image showing the tissues associated with the internal surface of the abdominal cuticle. **d** GFP is expressed in the abdominal body wall muscles and alary muscles of all abdominal segments (segments A3–A5 are shown). GFP expression was not detected in the oenocytes. The ventral surface abdomen was bisected (located on the lateral edges of the image) to expose the internal surface of the dorsal cuticle. The same preparation is shown in **c** and **d**. Reporter: UAS-StingerII. **e** Fluorescent image of a fillet preparation of a 3rd instar larval cuticle showing the body wall muscles. Reporter: UAS-mCD8-GFP. For each preparation, $n \geq 2$.

circuit that includes neurons associated with sensory processing, sensory-motor integration and motor output.

### *dokb^{+1}* and *dokb^{+2}* are natural variants

We have identified two alleles of *dokb* associated with distinct laboratory wild-type strains. We asked whether these two alleles are an artefact of captivity or, alternatively, may represent naturally occurring variants. The identification of such allelic variation in the wild could indicate an adaptive advantage. To investigate whether these allelic differences are found in nature, we examined the *dokb* sequence of various wild-caught *D. melanogaster* strains from PopFly[40] (Supplementary Fig. 10a). We analysed the frequency of cytosine and adenine at *dokb*'s 1049th nucleotide, which accounts for the alanine to glutamic acid change in our CS and OR laboratory strains, respectively. We found that adenine occurs at this position in 6.5% of the samples analysed from 30 strains taken from five continents around the world (Supplementary Table 3). In addition, we found that the distribution of

adenine across the different strain samples is unevenly distributed [$X^2(28, N = 49) = 369.16$, $p < 0.001$], with a higher frequency of adenine occurring in various strains collected from lower elevations (Supplementary Fig. 10b). Based on these observations, we conclude that these *dokb* alleles exist in the wild and show that the laboratory strains used in our experiments retain this natural variation.

## Discussion

By taking a social network approach to study groups of flies, we and others have characterized differences in group structure across strains within the *melanogaster* species[2]. There are many examples of strain-based distinctions in *Drosophila*, encompassing diversity in mating strategies[41], pheromonal profiles[42–44], biological clocks[45], taste reception[46], and learning abilities[47,48]. Our paper shows that a single gene, *dokb*, is sufficient to regulate a specific behavioural feature of social networks, betweenness centrality. Allelic differences of *dokb* within laboratory strains modulate betweenness centrality. In addition, we found identical alleles in wild populations, with the *dokb^{+2}* allele occurring at a relatively low frequency. Other low-frequency alleles have been reported to confer an adaptive advantage in certain populations. For example, the sickle haemoglobin (HbS) allele, a structural variant of normal haemoglobin, occur with varying frequencies in different populations, with higher HbS frequencies in regions with higher instances of malaria[49]. Similarly, we speculate that these two *dokb* alleles confer an adaptive value of social structure within the species and that, given the unequal distribution of the two alleles across elevation, the population benefits from having higher frequencies of the *dokb^{+2}* allele at lower elevations, although further investigation is required to determine what that advantage may be. Identifying this genetic variant is invaluable for investigating the evolutionary processes underlying group behaviour.

Although group-level social behaviour is a general feature of animal life, the molecular evolution of social networks and collective behaviours have not been well characterized. Surprisingly, we found no DNA or protein sequence homology of *dokb* outside of the Diptera order, nor did we find any conserved domains within the predicted CS and OR DOKB protein sequences. Nevertheless, in instances of convergent evolution, conservation often manifests at the pathway level rather than through identical gene sequences. For example, caste phenotypes in eusocial insects, including bees, ants, and wasps, have been associated with conserved metabolic pathways, such as the glycolysis pathway, rather than specific genes[50]. We anticipate that the molecular pathways and/or functional cell circuitry with which *dokb* is involved will be conserved in other animals. Numerous studies suggest genetic contributions to the structure of social networks[10–12], including for betweenness centrality in humans[13], indicating the potential for a conserved pathway. The functional roles and biological pathways associated with *dokb* and its encoded protein can now be directly addressed.

## Methods
### Fly Rearing
All fly strains were reared on a medium containing agar, glucose, sucrose, yeast, cornmeal, wheat germ, soya flour, molasses, propionic acid and Tegosept in a 12:12 h light/dark cycle at 25 °C. For network experiments, newly-eclosed adult males were collected using $CO_2$ anaesthesia and were kept in same-sex groups of 12–16 flies in food vials and aged for 3 days. Experiments were run between 9 and 10.5 h after the lights were on.

### Fly Lines
Canton-S (CS) and Oregon-R (OR) wild-type flies were obtained from J. C. Hall (Emeritus at Brandeis University, Waltham, MA). Introgression and recombinant lines were generated through a series of crosses from these wild-type lines. Recombinant lines had OR as their X and 2nd

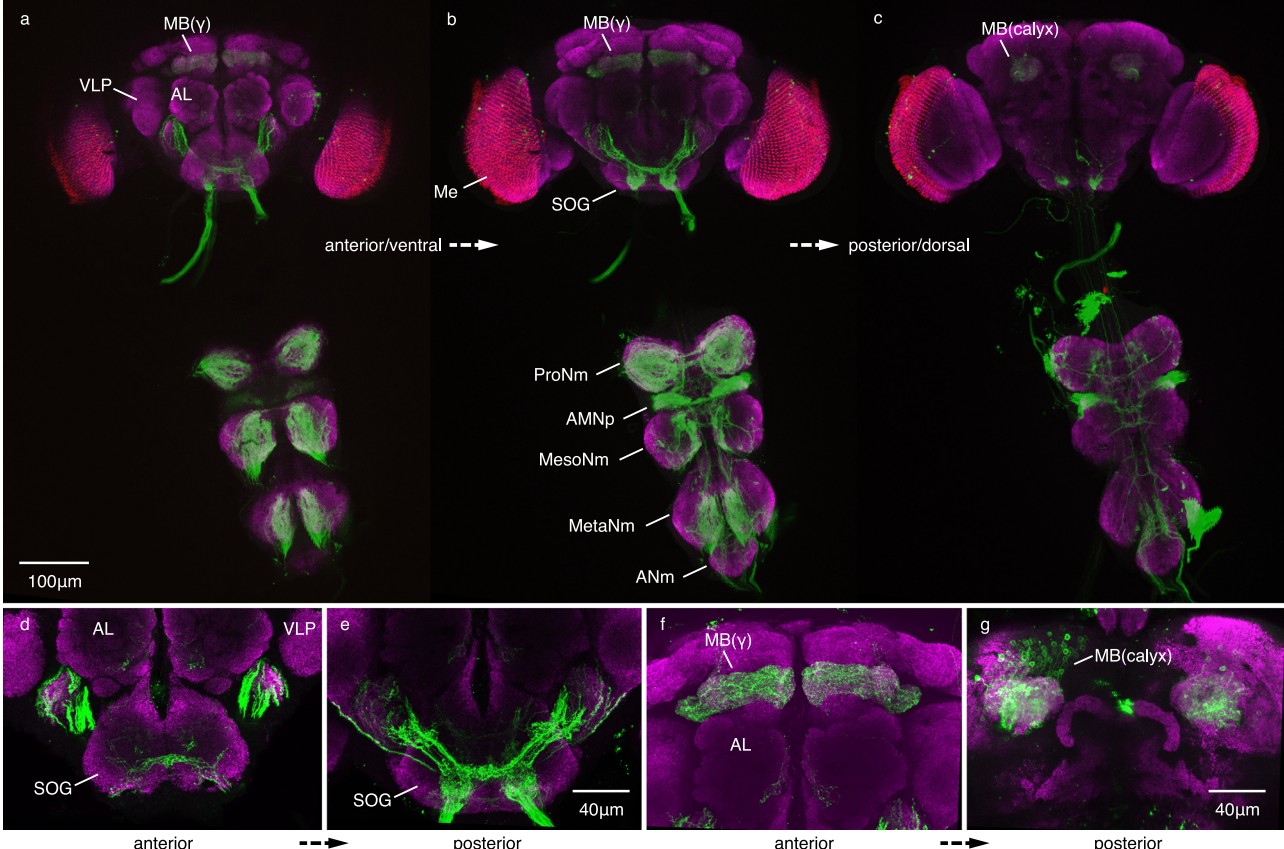

**Fig. 6 | Expression of *dokb* using *dokb^n2*-GAL4 line and a GFP reporter in the adult CNS.** *dokb^n2*-GAL4 expression patterns reported by UAS-mCD8.GFP (green). Immunostained with anti-nc82 antibody (magenta). Scale bars are indicated at the bottom. **a**–**c** Z-progression through the brain (anterior to posterior) and ventral nerve cord (ventral to dorsal). **d** and **e** Z-progression through the AL, VLP and SOG (anterior to posterior). Note the staining in the ventral-medial glomeruli of the AL. **f** and **g** Z-progression through the MB γ lobes (**f**) and calyces (**g**). MB mushroom body, VLP ventrolateral protocerebrum, AL antennal lobe, Me medulla, SOG suboesophageal ganglion, ProNm prothoracic neuromere, AMNp accessory mesothoracic neuropil, MesoNm mesothoracic neuromere, MetaNm metathoracic neuromere. *n* ≥ 5.

chromosome and an OR recombined with CS as their 3rd chromosome.

+;+;*dokb^n2*-GAL4, *dokb^n1*, CS(*dokb^{+2}*), *dokb^n2* and OR(*dokb^{+1}*) were generated using CRISPR/Cas9 in collaboration with WellGenetics Inc. Lines were verified with DNA sequencing.

*dokb* null lines (*dokb^n1*, *dokb^n2* and *dokb^n2*-GAL4) were generated such that the 980 bp coding sequence (+16 nt from ATG to −85 nt from stop codon of *dokb*) was deleted and replaced with a Stop-RFP cassette with 3-frames of stop codons and a 3XP3-RFP, except for the *dokb^n2*-GAL4 line which was replaced by a T2A-GAL4::VP16 cassette and 3XP3-RFP.

For the CS(*dokb^{+2}*) and OR(*dokb^{+1}*) swap lines, the entire gene region of *dokb*, from the promoter to the 3'UTR (1960bp) in one strain, was deleted and replaced by the gene region from the other strain with an inverted PBacDsRed marker inserted into the 2nd intron of *dokb*. The marker was excised before lines were used in behavioural experiments.

The *CG14109* deficiency null was generated by selecting progeny from crossing virgin +;+;*dokb^n2*-GAL4 females to deficiency males (w[1118]; Df(3L)ED4502, P{w[+mW.Scer\FRT.hs3] = 3'.RS5 + 3.3'} ED4502/TM6C, cu[1] Sb[1]; Bloomington Stock Center Line #8097).

The *Nplp2* RNAi knockdown flies were generated by selecting progeny from crossing *Nplp2*-GAL4 line (w[1118] (I); P{w[+mC] = Nplp2-GAL4}vie72a (II); Korean *Drosophila* Resource Center Line #10023) to the UAS-*Nplp2* RNAi line (y[1] v[1]; P{y[+t7.7] v[+t1.8] = TRiP.HMJ21484} attP40; Bloomington Stock Center Line #54041).

## Genetic mapping

To map the locus containing the gene responsible for regulating BC in flies, we based our method on Chen et al.[15] but adapted their method to suit our assay and fly strain requirements. We began by assessing the BC of our introgression lines and determined that there was a gene(s) on the 3rd chromosome responsible for regulating BC (Supplementary Fig. 2a). Next, we sequenced the genomes of our CS and OR lab strains: ~3 μg of gDNA was collected from male and female CS and OR whole flies using Zymo Research ZR Tissue and Insect DNA Miniprep Kit (#D6016) and supplied to The Centre for Applied Genomics (TCAG; Toronto, ON, Canada) for high-throughput sequencing (Illumina HiSeq 2500).

We subsequently generated a high-resolution SNP map [Clinical Genomics Centre (Toronto, ON, Canada)] by identifying polymorphisms between the two strains at a resolution of approximately 60 equally spaced SNPs on each chromosome 3 arm.

We generated recombinant lines through a series of genetic crosses that resulted in flies that had an OR X and 2nd chromosome and a recombined CS/OR 3rd chromosome. Instead of phenotyping our recombinant lines first [as was done in Chen et al.[15]], we genotyped our recombinants to determine which regions of the chromosome were OR and which were CS. The gDNA samples of recombinant lines were supplied to the Clinical Genomics Centre (Toronto, ON, Canada) for genotyping.

Then, we strategically chose lines to phenotype based on their recombination events. If the BC phenotype of a given recombinant line

was statistically the same as the CS control, we concluded that the CS region of that 3rd chromosome housed the gene(s) responsible for regulating BC and, alternatively, if the BC phenotype of a given recombinant line was statistically the same as the OR control, we concluded that the OR region of that 3rd chromosome housed the gene(s) responsible for regulating BC. Using this method, we narrowed the locus to a ~1.2 MB region (Fig. 2a).

We further narrowed and mapped the locus using SNPs within this 1.2 MB region and in-house PCR to genotype more recombinant lines. Ultimately, using this method, we narrowed the locus to 474 kb (Fig. 2b).

### Anti-*dokb* antibodies

Antibodies were raised in New Zealand rabbits against the synthesized peptide VRQSTEEEEVQSHV, which corresponds to amino acid positions 61-74 in the CG14109 protein. The resulting antibody preparation was termed anti-CG14109_AA_61-74. All peptide synthesis and antibody production were performed by GenScript (Piscataway, NJ, USA) using the PolyExpress polyclonal antibody express service.

### Protein extraction and Western Blot

Proteins from 15 male and 15 female fly heads per genotype were homogenized in lysis buffer, incubated on ice for 1 h, and centrifuged at 4 °C for 10 min. The supernatant was mixed with 2× Laemmli Sample Buffer (Bio-Rad, #1610737) and placed in a heating block (110 °C) for 15 min. Samples were loaded into a 4–20% Mini-PROTEAN® TGX™ Precast Protein Gel (Bio-Rad, #4561094). Proteins were transferred to a PVDF membrane using the Trans-Blot Turbo Transfer System (Bio-Rad). Blocking was performed using a 5% milk solution at 4 °C overnight. The membrane was incubated in a 5% milk solution + anti-CG14109_AA_61-74 antibody (1:1000) for 2 h at RT with gentle agitation. The membrane was washed three times using 1X PBS for 5 min. then incubated with a 5% milk solution + secondary antibody for 1 h at RT with gentle agitation. The membrane was washed three times using 1X PBS for 5 min and was subsequently exposed to Clarity Western ECL Substrates (Bio-Rad, #1705061) and imaged using a ChemiDoc Imaging Station (Bio-Rad).

### Social networks

Twelve male flies were gently aspirated into a circular plexiglass arena (60 mm diameter, 3 mm depth) and covered with a glass lid. The arena was placed under a FireflyMV camera (Point Gray) with an infra-red backlight in a 25 °C environmental chamber at 60% humidity. Flies were allowed to acclimate in the arena for 10 min. undisturbed. The subsequent 30 min. were recorded using fview[51]. The 30 min video was processed using Ctrax (v 0.5.13)[52] to determine the position and orientation of each fly. Each video was manually inspected, and tracking errors were corrected for each video using fixerrors[52]. Approximately 20 videos were used per genotype.

The social interaction criteria for all gene mapping experiments (introgression and recombinant experiments) were the same as the criteria used in Schneider et al.[16], and are described as follows:

(a) the angle subtended by the long axis of the interactor fly and the line segment connecting the interactor fly's center of area to that of the interactee fly is less than or equal to 90°

(b) the length of that line segment is less than or equal to two body lengths of the interactor fly

(c) these two conditions are maintained for at least 1.5 s

In 2014, a new method was proposed by Schneider and Levine to objectively and computationally determine the interaction criteria for a given genotype. Using such methods allows researchers to assess social networks while controlling for the different ways that a genotype may interact. After the software's creation, we used this method to determine the interaction criteria for all subsequent experiments for all other genotypes (excludes recombinant and introgression experiments; see Supplementary Table 4)[53].

Directed social networks were generated by calculating iterative networks using a moving-window boxcar filter at 25% network density (33 unique interactions), as detailed in Schneider et al.[16,54]. An interaction was considered unique if the interactor had not previously interacted with the other fly. The first network iteration represents the first 33 unique interactions, the second network iteration ignores the first unique interaction and adds a subsequent unique interaction…the $i$th network iteration ignores the first $i-1$ interactions. For $n$ network iteration, the betweenness centrality ($BC_n$) was determined by averaging the BC of all twelve flies. Betweenness centrality for each node (fly) was determined using the Brain Connectivity Toolbox MATLAB (Mathworks, v2014a) scripts[55], and was calculated using Eq. (1):

$$BC(\nu) = \sum_{s,t:s\neq\nu\neq t} \frac{\lambda_{st}(\nu)}{\lambda_{st}} \qquad (1)$$

where the BC for node $\nu$ [$BC(\nu)$] is the sum of the total number of shortest paths between nodes $s$ and $t$ that pass through $\nu$ [$\lambda_{st}(\nu)$] divided by the number of shortest paths from nodes $s$ to $t$ [$\lambda_{st}$]. Each network iteration's average BC was standardized for degree distribution. For each iteration, 10,000 random networks with the same in- and out-degree distribution were generated and each random network's BC was calculated. A $z$ score was determined using Eq. (2):

$$\frac{BC_{observed} - mean(BC_{random})}{std(BC_{random})} \qquad (2)$$

The BC $z$ scores for all iterations from a single 30 min. network was averaged to generate a mean BC for a single experiment, represented as a single dot in a dot plot graph showing betweenness centrality. In other words, each network (or $n=1$) indicates the mean $z$ score BC from an independent group of 12 flies that were discarded after the 30-min network experiment was acquired. For each experimental group, we acquired videos from ~20 independent groups of flies ($n=$ ~20 for each treatment).

For networks involving the CS{$dokb^{+2}$} and OR{$dokb^{+1}$} flies and their controls, movement (mm/s) and interaction rate (# interactions/min.) were calculated, along with three additional network properties: assortativity, clustering coefficient and global efficiency. All network properties were calculated and standardized as described above and in Schneider et al.[16].

### qPCR

To determine whether *dokb* expression correlates to BC, RNA was extracted from whole male flies using RNeasy Micro kit (Qiagen). cDNA was synthesized using iScript cDNA Synthesis Kit (Bio-Rad) and RT-PCR was performed using a CFX384 Real-Time System (Bio-Rad) and iTaq Universal SYBR Green Supermix (Bio-Rad). Each experiment was performed three times (biological replicates) and contained three technical replicates/samples. The reference gene used to normalize each sample was RpL32.

To detect whether *dokb* was expressed in pheromone-synthesizing cells, oenocytes were dissected from 40 w[1118]; nSyb-GAL4/+; +/+ flies (aged to 10 days) and pooled into a single biological sample. RNA extraction, cDNA synthesis and qPCR methods were as described above.

Primer sequences for detecting levels of *dokb* are as follows: 5′ GCGAGCGCCTAGCTGC and 5′ ATTCTCTTCTTGGGCACTCTCTACGG. Primer sequences for detecting levels of *Nplp2* are as follows: 5′ ATGGCCAAGCTCGCAATTTG and 5′ GTTGAAATCACCCTGGGCCT.

## RNAseq

CNS dissections and extractions were performed three times to produce three biological replicates. The protocol was as follows: CS and OR male flies housed in 12:12 h light/dark cycle at 25 °C in groups of 12 male individuals/vial. Three CNS dissections were performed over four time points for each genotype (total: 12 CNS/genotype). The CNS samples for each genotype were pooled. RNA was extracted using Qiagen RNeasy Microkit. RNA sequencing was performed on Illumina HiSeq 2000 by VANderbilt Technologies for Advanced Genomics (VANTAGE, Nashville, TN). RNA samples were prepared using Illumina TruSeq Stranded mRNA-seq kit and sequenced at paired-end 50 bp with a target of ~25 M reads/sample. The raw RNAseq data have been deposited in the NCBI's SRA database under accession code PRJNA1082663.

## Cuticular hydrocarbon extraction

Flies were sedated on ice and a single fly was added to a microvial containing 50 μl of hexane containing 10 ng/μl of octadecane (C18) and 10 ng/μl of hexacosane (nC26) as injection standards. Tubes were gently vortexed for 2 min. Flies were removed and samples were stored at −20 °C prior to analysis. Extracts that were obtained from 40 to 50 individuals per genotype and examined by gas chromatography.

## Immunohistochemistry

For detecting *dokb*-GAL4 expression across larval development in a central brain lobe, appropriately aged larval CNS samples were fixed with 4% paraformaldehyde for 20 min. at RT. The 1° antibody, rabbit anti-GFP (Cell Signaling Technology, #2956), was applied at 1:1000 and the counterstaining antibody, mouse anti-DN-cad (Cell Signaling Technology, #14215), was applied at a 1:5 concentration. Alex Fluor 488 Donkey Anti-Rb IgG (H + L) (Invitrogen, #A32790) and Alex Fluor 555 Donkey Anti-Rb IgG (H + L) (Invitrogen, # A31572) were applied at a 1:500 concentration.

For detecting *dokb*-GAL4 expression in the larval and adult CNS, samples were fixed in 4% paraformaldehyde for 20 min. at RT. Samples were labelled with rabbit anti-GFP.Alexa 488 conjugate (1:400; Invitrogen, # A21311) and counterstained with primary mouse anti-Brp (nc82) (1:40; DSHB) and secondary donkey anti-mouse Alexa 647 (1:400; Invitrogen, #A31571). Wandering 3rd instar male larvae were used. Adult male flies were aged 3-4 days old.

All images were obtained from a Zeiss LSM880 microscope.

## Imaging of muscular tissue

For imaging the adult thorax and abdomen, a UAS-nuclear GFP reporter was used (UAS-StingerII[56]. The reporter line was crossed to the *dokb*[n2]-GAL4 line. Male progeny were collected, and their abdomens were filleted for imaging. For larval images, a UAS-mCD8.GFP reporter was used, and 3rd instar larvae were filleted to expose the abdominal wall muscles. In the adult and larval muscle, expressed GFP was directly imaged. Images were obtained using a Zeiss Stereo Discovery V.12 microscope.

## Sequence homology searches

The DNA and protein sequences of *dokb* were retrieved from www.flybase.com using *CG14109* as the query. BlastN and BlastP searches were performed at NCBI using default setting. Conserved domain searches for the CS and OR DOKB putative protein were performed at https://www.ncbi.nlm.nih.gov/Structure/cdd/wrpsb.cgi using default search settings.

## Natural variant data and analyses

966 sequences of *Drosophila melanogaster* from 30 localities were downloaded from PopFly[40]. 215 sequences were removed due to nucleotide ambiguity of *dokb*'s 1049[th] nucleotide position. The frequency of each nucleotide at this position was manually tallied.

The world map depicting the various locations from which each strain was collected and created in MATLAB (MathWorks, v2023b). The longitude and latitude data for each location used to create the map and the elevation of each locality was downloaded from PopFly[40].

The chi-square test was performed manually in Microsoft Excel.

## Statistical analyses

ANOVA and *T*-tests were performed using MATLAB (MathWorks, v2014a). For all network experiments, outliers ≥75th quartile + (1.5 × IQR) or ≤ 25th quartile−(1.5 × IQR) were removed before statistical testing. All network experiments were statistically analysed with α = 0.05, except for the *dokb* swap experiment (Fig. 2c, d), where multiple measurements were assessed using the same data set (Bonferroni correction such that α = 0.008) and the hydrocarbon experiment (Supplementary Fig. 7, Bonferroni correction such that α = 0.01.)

## Reporting summary

Further information on research design is available in the Nature Portfolio Reporting Summary linked to this article.

## Data availability

Source data is available as a source data file. This file contains all behavioural, hydrocarbon and qPCR data generated in this study. The raw RNAseq data have been deposited in the NCBI's SRA database under accession code PRJNA1082663 Natural variant data from strains of *Drosophila melanogaster* were acquired from the PopFly database (https://popfly.uab.cat/)[40]. Source data are provided in this paper.

## Code availability

Code is available upon request.

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

## Acknowledgements

These studies were funded by grants to J.D.L. from NSERC (RGPIN-2022-04765), CIHR (MOP-133659), CRC, and CIFAR. The authors wish to thank the following people for their help, comments and feedback: Josh Dubnau, Ralph Greenspan, Daniel Kronauer, Gene Robinson,

Jean-Christophe Billeter, Michael Dickinson, Paul Taghert, Paul Breslin, Jacob Jezovit, Urfa Arain and Marla Sokolowski. The authors are especially thankful to Dr. Barbadilla and Dr. Hervás for their help and support with PopFly.

## Author contributions

R.R. and J.D.L. wrote the manuscript. J.J.K. and M.S. provided stained images of the muscle and CNS. R.R., M.G. and A.R. did social network experiments. R.R., J.S. and J.D.L. designed the experiments. J.D.L. supervised the work.

## Competing interests

The authors declare no competing interests.
