## [Peer Review File · Nature Communications]

The Gene “degrees of kevin bacon” (dokb) Regulates a Social Network Behaviour in *Drosophila melanogaster*REVIEWER COMMENTS

Reviewer #1 (Remarks to the Author):

This is a marvelous study. It is unique in behavior genetics in addressing a phenotype that is more typically studied in behavioral ecology. One of the challenges of behavior genetics is in defining a phenotype that can be measured and not just defined colloquially or subjectively. Traits like foraging, which is surely complex and involves many different behavioral acts, must be reduced to measures such as path length with an assumption that this measure accurately captures what people assume when reading a descriptor like foraging. Even more fraught is measuring traits that require social interactions to be expressed, such as dominance or parenting. How can this interaction be ascribed to an individual? Rooke et al. tackle this most difficult and subjective behavior, social interaction, using the complex measurement that has revolutionized the study of social interactions in nature – social network theory and one of its primary measures of “betweenness centrality”. Social network theory is rich, highly developed, and well calibrated and verified for measuring the importance of social interactions in a wide variety of organisms. Here, Rooke et al apply it to interactions amongst *Drosophila* and in a tour-de-force identify a gene that influences this measurement.

The work is pleasingly complete and the evidence robust with multiple strands. The power of *Drosophila* genetics and strains is used to great effect. It is hard to imagine how many more ways they could have investigated the role of *dokb*. That said, this work should lead to many additional related topics, such as how variation in the *dokb* of the group influences expression, how experience influences expression, etc.

As an aside, I love the name.

Line 5: change “Social networks represent interactions... and are...” to “Social networks are a mathematical representation of interactions..., which are...”

Line 33: “communication relay” – it feels like something is missing here. I’m not entirely sure what is meant by communication relay so I can’t suggest an alternative, although I’m assuming it is acting as a point in a chain of communication.

Line 125: it does seem sensical that sensory processing influences social interactions – does this relate to things like autism in humans?

Line 144: I agree it is likely to be a player in a pathway (see also arguments to this effect by Tim Linksvayer) , and identifying this pathway seems to me to be critical.

I would have Supplementary Figures 1 and 5 in the main manuscript.

Well done. This was a pleasure to review.

Allen Moore

Reviewer #2 (Remarks to the Author):

The manuscript “degrees of kevin bacon (dokb) Regulates a Social Network Behaviour in *Drosophila melanogaster*” reports on the discovery that the gene dokb contributes to the regulation of social network related behaviors by modulating the betweenness centrality of individuals within interacting groups of adult *Drosophila*. Understanding how social networks are regulated at the genetic and neuronal levels is originative, timely, and potentially important for understanding the role of sociality in physiology and behavior. However, in its current form, the study suffers from several weaknesses that somewhat reduce my overall enthusiasm for the project. Below are more specific comments that I hope the authors will find useful.

1. The direct relevance of the described studies and their interpretations to understanding how social networks are regulated in humans, or their overall relevance to health outcomes, are overstated and unnecessary.
2. No information is provided for the genetic and bioinformatic strategies used to quantitatively map SNPs associated with the focal phenotype.
3. The authors should provide more specific information for how No information should be provided for how “betweenness centrality” was calculated and analyzed.
4. In general, the authors provide very little information about how many of the complex data included in this study were evaluated for power or statistically analyzed.
5. Since the intergenic region between CG14109 and its downstream neighbor is less than 1Kb, the authors should also verify that the GAL4 KO allele does not affect its expression. The author should also show the full expression pattern of the GAL4 in the larval brain. This is particularly important for identifying its expression level in motor neurons, which could affect data interpretations.
6. Although the data for a causal relationship between SNPs in dokb and the regulation of social networks are solid, the argument that this effect is driven by its action in the nervous system is not as

well supported. In fact, the GAL4 data generated by the authors, and publicly available transcriptomic data indicate that the gene is highly enriched in somatic muscles. Since locomotion is a major factor that could affect observed level of social interactions, demonstrating that overall locomotion is normal in null animals is an important control.

7. Based on available RNA-seq data, *dokb* is a broadly expressed gene. Since the authors were able to show that they can effectively downregulate it by RNAi, a more comprehensive GAL4 screen could provide important mechanistic insights about how this gene might regulate social networks in *Drosophila*.

8. The data about global minor allele frequencies are interesting. Although the y-axis in Supp Fig. 10b is missing some values, these data indicate that the minor allele is extremely rare across all populations studied. Therefore, it is not clear what can be learned from these data in the context of the behavior described here.

Reviewer #3 (Remarks to the Author):

The vinegar fly, *Drosophila melanogaster*, exhibits complex and dynamic social structures that are modulated by the experience and composition of different individuals within the group. In addition, multiple measures of social network position have been shown to be heritable in *Drosophila melanogaster*.

However, what genes and genetic pathways influence these social network positions?

To begin to address this question, Rooke and colleagues took their previously published observation, which showed that different strains of *Drosophila melanogaster* behave differently in similar conditions; betweenness centrality is higher in the Canton-S (CS) strain compared to the Oregon-R (OR) strain of *Drosophila melanogaster*. These findings suggested different adaptations in the natural populations from which these strains originated. Moreover, the genetically tractable model organism, *Drosophila melanogaster*, facilitated a genetic dissection of these network structure differences. They used a novel recombinant mapping technique and exploited the behavioural differences between these strains of *Drosophila melanogaster* to identify a gene, CG14109, which they gave the moniker degrees of Kevin Bacon (*dokb*) after the parlour game “Six Degrees of Kevin Bacon”.

Manipulating the expression of *dokb*, using RNA interference and a CRISPR/Cas9 generated null mutant, resulted in networks with lower betweenness centrality than control animals. They then identified strain-specific alleles of *dokb*; the gene differs by seven nucleotides between the CS and OR strains, one of which results in a predicted amino acid difference (129 Ala-Glu), which they uncover is a naturally

occurring variant. By swapping the *dokb* allele between strains, they discovered that flies exhibited the betweenness centrality pattern dictated by the donor allele.

To allow for a detailed anatomical analysis of *dokb*-expressing cells, the authors used CRISPR/Cas9 genome editing to insert the GAL4 coding sequence into the endogenous gene to create a *dokb*-GAL4 allele. Using *dokb*-GAL4 to drive the expression of a membrane-bound fluorescent reporter revealed expression in the adult CNS and somatic muscles. The authors speculate on the significance of expression patterns on the dynamics of social network structure. However, this will require future experiments focused on the functional analysis of specific cell types and variants and is beyond the remit of this current manuscript.

This manuscript is a fascinating study. The findings are a first; although the structure of the social group has a genetic basis, this has been harder to demonstrate, and the authors have actually shown that the naturally occurring variation in the *dokb* gene can influence social network position and is potentially linked to different adaptations in the natural populations from which these strains originate.

This current study will facilitate a detailed mechanistic analysis of how genetic variation shapes the environments that individuals experience. In humans, genetic variation in social environment construction is hypothesised to influence behavioural development, including the development of mental illness, because genotypes differ in their likelihood of experiencing psychosocial stress and other risk factors for disease.

I only have a few minor comments/requests.

Minor requests

Could the authors speculate about how they might test the adaptive significance of these naturally occurring variants on behavioural development?

Does the *dokb*GAL4 reiterate endogenous *dokb* expression? Did the authors try their anti-*dokb* antibody by immunohistochemistry?

Line 12. 'governs' is a stretch—perhaps change to influences.

Line 40. “adapted recombinant mapping.” can you expand on what ‘adapted’ means in the text?

Line 101. GFP was detected in somatic muscle tissue, including the alary muscle. The authors should include the data for all the somatic muscles in the manuscript, preferably in the main text.

Generally, I would encourage the authors to expand on the materials and methods section “Social Networks”. It would help the non-aficionados understand how betweenness centrality is being measured in this manuscript.

Reviewer #4 (Remarks to the Author):

This manuscript characterizes a gene, named *dokb*, that influences social structure in flies. I think this is a really interesting direction of work, and the authors have performed some interesting characterizations of this gene. They are using a powerful system to study very complex social dynamics, which is exciting. However, I really struggled to understand the broader importance of the study, the hypotheses being tested, and many of the key methods.

The manuscript starts by discussing human social networks, noting that the authors have identified a gene that influences social networks in flies. Of course, it would be impossible to conduct a gene knockout study (etc) in humans, but I didn’t really understand the conceptual motivation. Why is this important, and what does it have to do with humans (if anything)? Why would finding a gene that influences human social networks be important?

Next, the manuscript indicates that an “adapted recombinant mapping technique” was used to identify two genes with mutations correlated with betweenness centrality. Despite reading the manuscript, supplementary materials, and figures, I still have no idea what this actually was. I assume this is something like QTL? If so, what about it is “adapted”? The sample sizes and analysis methods are only reported in a figure caption, which made the writing hard to follow. Moreover, the approach identified two loci – is that more than would be expected by chance in a fruit fly-sized genome? Only one of the loci seemed to actually be associated with the social network measurements – is a 50% success/failure rate appropriate for this methodology? Finding this gene is a core aspect of the manuscript so this lack of clarity was concerning.

I was also unclear exactly how the social networks were measured. The methods section reported that the networks were measured exactly as in a previous paper by the authors. I understand that there is

often severe space limitation in these kinds of journals, but since this is a fundamental part of the manuscript it would be nice to orient readers to the key methods. In particular, it seems important to discuss how long the animals were measured for and what interaction criteria were used. Were these networks of affiliative behavior, antagonistic behavior, proximity, or all of these? This is really important to interpreting the results. Similarly, the fact that only males were measured is buried in a figure caption. I'm assuming this is because the females aren't important to the study, but why not?

In part because of this confusion I was unsure how to interpret the finding that *dokb* "regulates" betweenness centrality. As described in the beginning of the manuscript, it seems like individuals with high betweenness centrality have more connections in the network, particularly between otherwise sparsely-connected subgroups or individuals. I'm not completely clear on how a network can have higher betweenness centrality on average. If the network has denser connections overall, it would seem to follow that no single individual is particularly central. If this is correct (and it may not be!) then how was betweenness centrality correlated with particular recombinant lines? I assume that each network contained individuals all from the same line, so how did this work?

Next the manuscript dives into functional characterization of the *dokb* which seemed relatively straightforward. I found it interesting that *dokb* is expressed at the larval stage and in the adult mushroom bodies. I was confused by the interpretation of the hydrocarbon results that "changes in hydrocarbon profiles in *dokb* null mutants are a consequence of their social experience." What social experience do did the flies have? Wouldn't this be a confound in all of the other functional studies as well?

The manuscript also pointedly describes the frequency of the identified *dokb* mutation in wild-collected flies. Again, it was unclear what this was supposed to demonstrate. There are many thousands of mutations segregating in natural populations, particularly for species like flies with very large population sizes. What is the null or alternative hypothesis here?

At the end of the manuscript the ideas about humans are raised again. Since *dokb* does not have a human homolog, the discussion section focuses on gene networks. While it's certainly reasonable to imagine that *dokb* acts within a network that has a human homolog, the sudden introduction of networks when the central hypothesis was not supported seemed very post hoc. Or, perhaps the initial hypothesis -- that individual genes could be identified in flies that may shed light on humans -- was a bit of a "straw man"? Either way, this was not an effective way to communicate the importance of the results, which I am still unsure about.

The discussion finishes with some ideas about evolution, citing *dokb* as potentially "contributing to the plasticity of social structure within the species." This idea seemed promising, but what plasticity has been characterized in this species? Is that plasticity relevant to betweenness centrality? Also, what does

plasticity mean here? Usually I think of plasticity as change with the environment, but here the manuscript focuses on genetic differences without asking whether or how these are modulated by the environment.

As a very minor note, the name “degrees of kevin bacon” is cute and relevant, but it can sometimes be dicey to name things after prominent people, especially if they are still alive.

Overall, I think this manuscript has some promise but there are serious issues with precision of ideas, hypotheses, and analyses, and it is as yet unclear how this work will contribute to a broader scientific field.

Dear editor and reviewers,

I wish to thank the reviewers for their excellent feedback and suggestions. We have written responses to the reviewers' comments below (in bold). Changes in the manuscript are highlighted in yellow (see main text document). For ease of reading, we included the supplementary data together with the main text as a single file.

I sincerely hope you enjoy our revised manuscript and deem it worthy of publication.

Sincerely,

Joel Levine

Reviewer #1 (Remarks to the Author)

*“This is a marvelous study. It is unique in behavior genetics in addressing a phenotype that is more typically studied in behavioral ecology. One of the challenges of behavior genetics is in defining a phenotype that can be measured and not just defined colloquially or subjectively. [...] Rooke et al. tackle this most difficult and subjective behavior, social interaction, using the complex measurement that has revolutionized the study of social interactions in nature – social network theory and one of its primary measures of “betweenness centrality”. Social network theory is rich, highly developed, and well calibrated and verified for measuring the importance of social interactions in a wide variety of organisms. Here, Rooke et al apply it to interactions amongst *Drosophila* and in a tour-de-force identify a gene that influences this measurement.*

*The work is pleasingly complete and the evidence robust with multiple strands. The power of *Drosophila* genetics and strains is used to great effect. It is hard to imagine how many more ways they could have investigated the role of *dokb*. That said, this work should lead to many additional related topics, such as how variation in the *dokb* of the group influences expression, how experience influences expression, etc.*

As an aside, I love the name.”

Thank you Dr. Moore!

Line 5: change “Social networks represent interactions... and are...” to “Social networks are a mathematical representation of interactions..., which are...”

We changed the wording of this sentence per the reviewer’s suggestion and thank the reviewer for their feedback.

Line 33: “communication relay” – it feels like something is missing here. I’m not entirely sure what is meant by communication relay so I can’t suggest an alternative, although I’m assuming it is acting as a point in a chain of communication.

To help the reader understand how betweenness centrality works, we added a new Figure 1 (see revised manuscript) and expanded on its definition (see lines 34-36).

Line 125: it does seem sensical that sensory processing influences social interactions – does this relate to things like autism in humans?

We also see this connection but, given that other reviewers have requested we dial-back the relationships between our work and human biology, so we have decided not to elaborate in the manuscript.

Line 144: I agree it is likely to be a player in a pathway (see also arguments to this effect by Tim Linksvayer), and identifying this pathway seems to me to be critical.

Agreed. It's hard to imagine any behavioural phenotype that relies exclusively on only one gene. We are currently working to identify the biochemical and cellular pathways that include *dokb* and contribute to betweenness centrality. These results will be presented in a future study.

I would have Supplementary Figures 1 and 5 in the main manuscript.

Thank you for this suggestion. We moved Supplementary Figure 5 (now Figure 3) to the main manuscript. However, Figure 2a,b (formerly Figure 1) is a summary of the data presented in Supplementary Figure 2 (formerly Supplementary Figure 1). For this reason, we chose to keep Supplementary Figure 2 in the supplementary data so as not to be redundant. Nevertheless, if you or the editor want us to move Supplementary Figure 2 into the main text, we are happy to do so.

Well done. This was a pleasure to review.

Allen Moore

Reviewer #2 (Remarks to the Author):

“The manuscript “degrees of kevin bacon (dokb) Regulates a Social Network Behaviour in Drosophila melanogaster” reports on the discovery that the gene dokb contributes to the regulation of social network related behaviors by modulating the betweenness centrality of individuals within interacting groups of adult Drosophila. Understanding how social networks are regulated at the genetic and neuronal levels is originative, timely, and potentially important for understanding the role of sociality in physiology and behavior.”

1. The direct relevance of the described studies and their interpretations to understanding how social networks are regulated in humans, or their overall relevance to health outcomes, are overstated and unnecessary.

We imagine that this reviewer, along with reviewer #4, is understandably concerned about an attempt to elevate the general importance of our work by suggesting that it is relevant to humans. However, there are two key considerations: 1) There is an extensive literature on human social networks and, as we note in the text, the use of social network analyses originated with humans. We draw on this literature to explain social network concepts and implications of different social structures. Ultimately, applying social network analyses to humans is not different from applying it to any other animal. 2) Humans are animals. The human genome lies on a continuum of evolutionary genomic data and our results in *Drosophila* add to help explain the genetic contributions to understand social organization. This is similar to what has been shown for the *per* gene in circadian clocks and the cAMP signalling pathway in learning. There have only been a handful of studies that have looked at the contribution of heritability to social network structure, and one of those studies was performed on human twin social networks (others, which we have also cited in the main text, include marmots, macaques and *Drosophila*). The word “human” appears three times (excluding references) in the manuscript. We think that our references to the human literature is rational and relevant.

If the editor or reviewer feel strongly about this, we can remove any reference to humans in the text, but we believe that this would provide an oddly biased view of the field.

2. No information is provided for the genetic and bioinformatic strategies used to quantitatively map SNPs associated with the focal phenotype.

We have expanded and elaborated on our methods for SNP mapping.

3. The authors should provide more specific information for how No information should be provided for how “betweenness centrality” was calculated and analyzed.

To address this, we expanded and elaborated on our methods for network analyses and the calculation of betweenness centrality. We now include the equation used to calculate betweenness centrality, as well as a new figure (Supplementary Figure 1) that explains how our networks were generated and betweenness centrality calculated.

4. In general, the authors provide very little information about how many of the complex data included in this study were evaluated for power or statistically analyzed.

We elaborated on our statistical analyses section in the methods and these methods are also published in other papers that we cite from our lab.

5. Since the intergenic region between *CG14109* and its downstream neighbor is less than 1Kb, the authors should also verify that the *GAL4 KO* allele does not affect its expression.

We are aware of the possibility that CRISPR/Cas9 might be associated with off target effects. We appreciate the reviewer's concern about the potential downstream effects of the *Gal4 KO* mutant. Unfortunately, the *Gal4 KO* mutant is homozygous lethal which is why we also used a different *KO* mutant in our manuscript, without the *Gal4* insertion. This *KO* mutant was actually used to create the *Gal4 KO* mutant line. Nevertheless, we looked at the downstream gene's expression (*combover*) in our two WT lines and the homozygous *dokb* knockout (without *Gal4* insertion). The qPCR data for *combover* expression in these lines is shown below. There is no significant difference in *combover* expression between these lines.

All of the target sites of our CRISPR/Cas9 lines were verified by sequencing. Although this data does not eliminate the possibility of off target effects, given that the evidence of our recombinant mapping experiments and our swap experiment point to *dokb* as the basis for our effects on betweenness centrality, we think that it is unnecessary to include these data, which focus on *combover*, in the manuscript.

If the editor or reviewer feel strongly about this, we would be happy to include these data in our paper.

The author should also show the full expression pattern of the *GAL4* in the larval brain. This is particularly important for identifying its expression level in motor neurons, which could affect data interpretations.

This now shown in Supplementary Figure 9.

6. Although the data for a causal relationship between SNPs in *dokb* and the regulation of social networks are solid, the argument that this effect is driven by its action in the nervous system is not as well supported. In fact, the GAL4 data generated by the authors, and publicly available transcriptomic data indicate that the gene is highly enriched in somatic muscles. Since locomotion is a major factor that could affect observed level of social interactions, demonstrating that overall locomotion is normal in null animals is an important control.

The reviewer makes two important points: the importance of the central nervous system over other tissues and the potential confounding effects of locomotor activity with betweenness centrality. Both of these issues were addressed explicitly in the first fly social network paper by Schneider, Dickinson & Levine (2012). There, we showed that the formation of social networks requires normal chemosensory and haptic input. The fact that our *dokb*-Gal4 line drives expression in the mushroom bodies, a part of the nervous system known to integrate such sensory inputs, strongly suggests that an emphasis on the nervous system is parsimonious. In that paper we also demonstrate that there is no statistical difference in movement in the two wildtype lines (CS and OR) but that CS flies form networks with higher betweenness centrality than OR. In addition, numerous subsequent studies in *Drosophila melanogaster* have shown a similar result:

Liu *et al.* (2018) showed that isolated flies walked significantly more than flies in control groups starting from day 2 of the experiment onwards. However, there were no significant differences in global network parameters (i.e. global efficiency, assortativity).

Rooke *et al.* (2020) shows mean movement does not significantly differ across three different group sizes and three different group densities of *Drosophila melanogaster*. However, global structures, such as clustering coefficient and betweenness centrality, increase as group size increases.

Looking at social network structure of various genotypes across nutritional environments, Wice & Saltz (2021) found that increased variation in how central individuals were to the structure of their social groups could not be attributed to differences in the overall locomotor activity of each fly.

Alwash *et al.* (2021) show that rover flies move significantly more than sitter flies but that betweenness centrality of rover flies do not differ from that of sitter flies.

We have plotted data from our current study to help illustrate the lack of correlation of movement with social network structure (see below). The figure shows movement and BC for two different experiments (the *Nplp2* knockdown experiment on the top and the *dokb* null experiment on the bottom). You can see on the top that *Nplp2* knockdown flies move significantly less than their controls, but form networks with the same BC as their controls. Conversely, the *dokb* null mutants move more than their controls but form networks with significantly lower BC.

We did not include this movement data in the manuscript, as this is well established in the literature. However, if the reviewer feels strongly, we can include this as a supplementary figure.

7. Based on available RNA-seq data, *dokb* is a broadly expressed gene. Since the authors were able to show that they can effectively downregulate it by RNAi, a more comprehensive GAL4 screen could provide important mechanistic insights about how this gene might regulate social networks in *Drosophila*.

We thank the reviewer for this comment and completely agree that a more comprehensive Gal4 screen likely would provide important mechanistic insights into *dokb* regulation of social networks. We are currently pursuing this avenue of research but, as noted by reviewer #3, think that it is beyond the scope of this manuscript.

8. The data about global minor allele frequencies are interesting. Although the y-axis in Supp Fig. 10b is missing some values, these data indicate that the minor allele is extremely rare across all populations studied. Therefore, it is not clear what can be learned from these data in the context of the behavior described here.

We thank the reviewer for pointing out the error on the y-axis in Supplementary Figure 10b. We have fixed the y-axis labelling.

Regarding the issue of allele frequencies, the reviewer is correct that we are not in a position to use these data to comment on betweenness centrality. In our manuscript, we now are more explicit with why we looked at *dokb* allele frequencies in natural populations and conclude that their presence is not specific to lab strains.

This reviewer's comment prompted us to research the importance of low frequency alleles and we found compelling and interesting data that we now cite in our discussion (please see lines 164-173). Thank you.

Reviewer #3 (Remarks to the Author):

*“[...] This manuscript is a fascinating study. The findings are a first; although the structure of the social group has a genetic basis, this has been harder to demonstrate, and the authors have actually shown that the naturally occurring variation in the *dokb* gene can influence social network position and is potentially linked to different adaptations in the natural populations from which these strains originate.*

This current study will facilitate a detailed mechanistic analysis of how genetic variation shapes the environments that individuals experience. In humans, genetic variation in social environment construction is hypothesised to influence behavioural development, including the development of mental illness, because genotypes differ in their likelihood of experiencing psychosocial stress and other risk factors for disease.

I only have a few minor comments/requests.”

Minor requests

Could the authors speculate about how they might test the adaptive significance of these naturally occurring variants on behavioural development?

We thank the reviewer for their interest and their insightful questions. Understanding the adaptive significance or fitness consequences of the different network structures is not something we have investigated yet. For example, we have not yet looked at what the structure of a mixed OR/CS group looks like, nor have we looked at networks from any of the PopFly strains we present in the manuscript. However, as was stated in the manuscript, in some networks having a high BC can be advantageous and in other networks having a low BC can be advantageous. We now speculate that two strains of the same species forming networks with different BC properties acts as a “safety net” for the species and must convey some advantage, given the two variants’ preservation in natural populations (see lines 164-173).

*Does the *dokbGAL4* reiterate endogenous *dokb* expression? Did the authors try their anti-*dokb* antibody by immunohistochemistry?*

We did create anti-*dokb* antibodies and are able to show a specific band present in WT flies compared to *dokb* null flies (Supplementary Figure 4), however these antibodies unfortunately failed to give us signal when used for immunohistochemistry.

Line 12. ‘governs’ is a stretch—perhaps change to influences.

Done.

Line 40. “adapted recombinant mapping.” can you expand on what ‘adapted’ means in the text?

Done. This was elaborated on this in the Methods section.

Line 101. GFP was detected in somatic muscle tissue, including the alary muscle. The authors

should include the data for all the somatic muscles in the manuscript, preferably in the main text.

Done. See our new Figure 5.

Generally, I would encourage the authors to expand on the materials and methods section "Social Networks". It would help the non-aficionados understand how betweenness centrality is being measured in this manuscript.

Done. We have expanded on our social network methods and included a new figure describing our social network analyses (see our new Supplementary Figure 1).

Reviewer #4 (Remarks to the Author):

*This manuscript characterizes a gene, named *dokb*, that influences social structure in flies. I think this is a really interesting direction of work, and the authors have performed some interesting characterizations of this gene. They are using a powerful system to study very complex social dynamics, which is exciting. [...]*

The manuscript starts by discussing human social networks, noting that the authors have identified a gene that influences social networks in flies. Of course, it would be impossible to conduct a gene knockout study (etc) in humans, but I didn't really understand the conceptual motivation. Why is this important, and what does it have to do with humans (if anything)? Why would finding a gene that influences human social networks be important?

In our manuscript, we call attention to a point that is widely held in the literature: that social networks are expressed in a broad spectrum of species, from flies to humans. Indeed, network organization has been increasingly useful for studying biological systems in general. As noted above in response to reviewer 2's comments, our discussion of human networks followed from the fact that these are studied so extensively.

The importance of finding a gene that regulates *any* network (from any species) is that it shows that a biological structure (i.e. genes, proteins) underlies group level behaviour and provides a way to understand how such group level behaviours may be inherited. This can have implications for the evolution of sociality. We are not sure if this answers the reviewer's question but would be happy to elaborate if need be.

Next, the manuscript indicates that an "adapted recombinant mapping technique" was used to identify two genes with mutations correlated with betweenness centrality. Despite reading the manuscript, supplementary materials, and figures, I still have no idea what this actually was. I assume this is something like QTL? If so, what about it is "adapted"?

We address this in our expanded methods section.

The sample sizes and analysis methods are only reported in a figure caption, which made the writing hard to follow.

We have expanded and elaborated our methods section to better address this, and other, reviewers' concern about our lack of clarity and specificity. We kept our sample sizes in our figure legends due to the fact that we are presenting a lot of experiments with varying sample sizes. This permits the reader to understand what is being graphed and how it is being analyzed more directly, on an experiment-to-experiment basis.

Moreover, the approach identified two loci – is that more than would be expected by chance in a fruit fly-sized genome? Only one of the loci seemed to actually be associated with the social network measurements – is a 50% success/failure rate appropriate for this methodology? Finding this gene is a core aspect of the manuscript so this lack of clarity was concerning.

We set out to investigate whether the difference between CS and OR networks relied on a genetic difference. We inferred that it did because the differences in their networks are inherited. When we set out to map the loci, we had no a priori expectations as to how

many loci we would find. With any genetic screening, whether that be via mutagenesis or mapping a natural variant, one might identify numerous loci (see the screen done to identify the genetic control of early embryonic development by the Nobel prize laureates for Medicine or Physiology in 1995) or only a few loci (see the screen used to identify the *per* gene by Konopka & Benzer in 1971). We do not expect that *dokb* is the only gene that regulates BC, only that it is one gene that does, and it happens to be the gene we mapped. We expect to use this gene to identify other genes, as well as the cells that are required, for the control of social networks.

I was also unclear exactly how the social networks were measured. The methods section reported that the networks were measured exactly as in a previous paper by the authors. I understand that there is often severe space limitation in these kinds of journals, but since this is a fundamental part of the manuscript it would be nice to orient readers to the key methods. In particular, it seems important to discuss how long the animals were measured for and what interaction criteria were used. Were these networks of affiliative behavior, antagonistic behavior, proximity, or all of these? This is really important to interpreting the results.

We elaborated on our methods section to address yours and the other reviewer's request for more information and included a new figure to better explain our protocol (see Supplementary Figure 1). We have also added a new Supplementary Table 3 to illustrate the different interaction criteria used for each line when analyzing their social networks. The social interactions analyzed in our networks were based on temporal and spatial proximity.

Similarly, the fact that only males were measured is buried in a figure caption. I'm assuming this is because the females aren't important to the study, but why not?

In an earlier paper [Schneider, Dickinson & Levine (2012)], we showed that the betweenness centrality of networks formed by males and females are the same, for both the CS and OR strains. Due to this result, we chose to examine only male networks as males are simply easier to use. Females lay eggs in our experimental arena, and it interferes with our machine vision tracking software. We have included that we use only males in our behavioural experiments in our methods section (see line 332).

As described in the beginning of the manuscript, it seems like individuals with high betweenness centrality have more connections in the network, particularly between otherwise sparsely-connected subgroups or individuals. I'm not completely clear on how a network can have higher betweenness centrality on average. If the network has denser connections overall, it would seem to follow that no single individual is particularly central. If this is correct (and it may not be!) then how was betweenness centrality correlated with particular recombinant lines? I assume that each network contained individuals all from the same line, so how did this work?

We thank the reviewer for their impressive insight into the potential correlation between number of connections (also known as degree) and betweenness centrality. It is true that these two variables are often correlated. However, a high betweenness centrality does not require a high number of connections. In an attempt to clarify what we mean by betweenness centrality and address the reviewer's intuition about number of connections, we created Figure 1. This figure demonstrates that an individual with high betweenness centrality may have fewer connections than other individuals in the network.

Regarding the reviewer's concern about how betweenness centrality is calculated and network density, we expanded our methods section and now summarize our analyses in Supplementary Figure 1.

*Next the manuscript dives into functional characterization of the *dokb* which seemed relatively straightforward. I found it interesting that *dokb* is expressed at the larval stage and in the adult mushroom bodies. I was confused by the interpretation of the hydrocarbon results that "changes in hydrocarbon profiles in *dokb* null mutants are a consequence of their social experience." What social experience do did the flies have?*

In this manuscript, when we say "social experience" we are referring to the differences in social network structure that the flies produce, as shown in our network experiment results. We attribute "social experience" to this because all fly lines/genotypes used in our behavioural experiments are reared, housed and manipulated identically, but their social networks are different. For the hydrocarbon experiment, within a strain, the genotypes are also identical except for the removal of the *dokb* gene.

We show that CS and OR flies form networks with significantly different BC. As discussed in Krupp et al. (2008) and Kent et al. (2008), flies of the same genotype express different hydrocarbon profiles based on the composition of their social group. In these studies, flies were placed in mixed groups for the duration of the experiment (but were not reared in mixed groups). Kent et al. (2007) showed that the hydrocarbon profile can change within half an hour, which is the duration of our network experiments. Given our previously published results and the data we show in our current manuscript, we conclude that the changes in hydrocarbon profiles that we see in *dokb* null flies are likely due to the changes in their deletion-induced social structure/experiences rather than from *dokb* directly affecting hydrocarbon production.

Although we have never directly tested the issue of causality, we favor the hypothesis that the social environment influences gene expression, physiology and behaviour, as indicated by Krupp et al. (2008). Social networks provide insight into the structure of the social environment. *dokb* influences their social structure. Consequently, we speculate that the influence of *dokb* stems from its affect on the social group, not from a direct regulatory effect of these other measures. We think that there is enough evidence in the literature to support our conclusion.

To better illustrate our rationale, we have added text to the hydrocarbon results section of our manuscript.

Wouldn't this be a confound in all of the other functional studies as well?

The relationship between hydrocarbon profiles and social experience acts as a feedback loop where changing one variable will/can affect the other [Krupp et al. (2008), Kent et al. (2008), Billeter et al. (2009)]. With our current study, we question whether *dokb*-related changes in social structure change hydrocarbon profiles or whether *dokb*-related changes in hydrocarbon profiles affect social structure. There is no evidence that *dokb* directly affects hydrocarbon synthesis or production simply because *dokb* is not expressed in the oenocytes, which are the hydrocarbon-producing cells in insects. Thus, because *dokb* likely does not directly affect hydrocarbon production, we conclude that

the changes in social network structures of *dokb* null flies likely results in altered hydrocarbon profiles.

We do not think that this is a confound in our experiments, but rather a result we are capturing.

The manuscript also pointedly describes the frequency of the identified dokb mutation in wild-collected flies. Again, it was unclear what this was supposed to demonstrate. There are many thousands of mutations segregating in natural populations, particularly for species like flies with very large population sizes. What is the null or alternative hypothesis here?

We thank the reviewer for their comment, which prompted us to include more rationale in our results section (see lines 144-147) and elaborate our discussion section (see lines 164-173).

Briefly, the CS and OR strains used in our lab (and hundreds of other fly labs) have been retained for thousands of generations in lab-based environments conducive to inbreeding. After identifying the *dokb* locus and its two alleles, we questioned whether these alleles were found in only our lab-based strains or persisted in wild populations. Thus, our null hypothesis was that these alleles were only present in inbred lab-based populations. To investigate, we used the Popfly database and discovered these two alleles in most of the wild populations we examined. We are drawing attention to a non-synonymous mutation and are intrigued by the presence of this mutation in so many wildtype strains. We think this could speak to an adaptive advantage provided by alternative social structures embedded within a population of flies.

We draw parallels between the *dokb* alleles and the mutation which gives rise to sickle hemoglobin (HbS, which causes sickle cell disease). In both instances, there is a single amino acid difference between alleles. In addition, HbS confers a survival advantage in some environments, and we predict that the *dokb* alleles do the same (although we do not know *which* environments).

At the end of the manuscript the ideas about humans are raised again. Since dokb does not have a human homolog, the discussion section focuses on gene networks. While it's certainly reasonable to imagine that dokb acts within a network that has a human homolog, the sudden introduction of networks when the central hypothesis was not supported seemed very post hoc. Or, perhaps the initial hypothesis -- that individual genes could be identified in flies that may shed light on humans -- was a bit of a "straw man"? Either way, this was not an effective way to communicate the importance of the results, which I am still unsure about.

This reviewer shares a concern stated above by reviewer # 2 (please see our response above). The fact is that humans are one of the most studied animals from a social network perspective. Fowler *et al.* (2009) proposed that there was likely a genetic component to human social networks and found that BC was likely heritable in humans. To date, no one in any animal system has identified a gene that regulates social networks. With this in mind, the identification of *dokb* offers an entry point toward understanding the relationship between social network structure and inheritance. Further, as cited in our discussion, conserved pathways are known to underlie social systems in eusocial insects. Consequently, we think that *dokb* provides insight into the

genetic organization of social behaviour and might be associated with a conserved pathway across several animals. This sort of speculation seems reasonable to us.

However, we have addressed this comment by:

1. Removing “conserved pathways from flies to primates” from our discussion.
2. Emphasizing the role of *dokb* and its associated pathway in the inheritance of social structure.

We appreciate the author’s concern that the initial mention of a single gene might have been a “straw man”. This is not the case. Rather, we assume that the identification of a single gene would necessarily lead to the identification and understanding of the role gene pathways play in social structure. As we have mentioned in response to reviewers above, it is highly unlikely that a single gene would be solely responsible for this, or any, behaviour.

The discussion finishes with some ideas about evolution, citing dokb as potentially “contributing to the plasticity of social structure within the species.” This idea seemed promising, but what plasticity has been characterized in this species? Is that plasticity relevant to betweenness centrality? Also, what does plasticity mean here? Usually I think of plasticity as change with the environment, but here the manuscript focuses on genetic differences without asking whether or how these are modulated by the environment.

There are many examples of plasticity in *Drosophila* (a review was just published on this topic by Chen & Sokolowski, 2022). However, considering your comment and in an effort to be clear, we changed the text in our discussion by replacing “plasticity” with “adaptive value”.

As a very minor note, the name “degrees of kevin bacon” is cute and relevant, but it can sometimes be dicey to name things after prominent people, especially if they are still alive.

We have contacted Kevin Bacon and received written permission to name our gene “Degrees of Kevin Bacon”

REVIEWERS' COMMENTS

Reviewer #2 (Remarks to the Author):

Overall, the revised manuscript “degrees of kevin bacon (dokb) Regulates a Social Network Behaviour in *Drosophila melanogaster*” is significantly improved.

As the authors clearly state in their response, social networks are likely to drive behavior in all animals, including humans. However, my original criticism is not about the general applicability of social network analysis to behavior in humans, flies, or any other animal species. Instead, what I was trying to argue is that at the level of genetics, the described findings are not necessarily directly relevant to human biology. All that can be said based on the included data is that the outcomes of this study in the fly may represent a conceptual mechanistic framework that is analogous to the neurogenetic processes that drive similar behavioral interactions in humans. Regardless, I'll leave the decision on how strongly the human connection should be emphasized to the authors and the editor.

Otherwise, the authors did an excellent job addressing all other major issues raised by the reviewers. It is a very nice paper. Congratulations.

Reviewer #3 (Remarks to the Author):

The authors have responded to all the acceptable revisions/requests outlined by the reviewers. The paper is much improved, and the authors are to be applauded for their efforts.

Reviewer #4 (Remarks to the Author):

The authors have done a great job of substantially clarifying the manuscript, and as a result the manuscript is much improved and many of my concerns have now been addressed. I have two major concerns that should be straightforward to address, and a few lingering minor concerns. Overall I am pleased with the revisions and think the manuscript is nearing readiness for publication.

Major concern:

-I appreciate that the relevance of natural allelic variation has been clarified. However, some of the new sections add confusion, specifically:

(line 146) "The identification of such allelic variation in the wild could indicate an adaptive advantage."

And

(line 170) "Similarly, we speculate that these two dokb alleles confer an adaptive value of social structure within the species and that, given the unequal distribution of the two alleles across elevation, the population benefits from having higher frequencies of the dokb+2 allele at lower elevations, although further investigation is required to determine what that advantage may be."

These sentences incorrectly imply that the fitness consequences of dokb occur at the population or species level, or that the mere presence of an allele in a natural population indicates that the allele is advantageous. I strenuously oppose the idea that a *population* benefits from having a particular allele frequency (without direct evidence). In addition, we don't yet know that dokb alleles have any phenotypic effect in wild populations, or that the elevational gradient is due to direct selection rather than linkage or population structure.

A clearer summary of this evidence would be along the lines of "our findings suggest that these alleles of dokb experience different selection pressures at different elevations, although further investigation is required to determine what the sources of selection, if any, may be."

-I still think that the manuscript does not sufficiently explain what it means for an entire network to have lower or higher betweenness centrality. Figure 1 just adds to this confusion by only explaining BC as an individual attribute. Can you illustrate or explain BC at a network level? Maybe a sentence along the lines of "briefly, a network with high BC has [X qualitative characteristics] while a network with low BC is characterized by [the opposite]"

Minor concerns

-I still think the importance of this research direction is not fully explained in the introduction (how do these results change our thinking about networks, compared to other results that the authors could have obtained?), but the introduction is at least clear and accurate now which is great.

-The section on recombination mapping is much clearer, so I now understand much more of what was done, thank you! To clarify, was only the 3rd chromosome tested? If so, that's fine, but it might be good to make this context more explicit.

REDACTED